# Suckling Behaviour of Beef Calves during the First Five Days Postpartum

**Lindsay A. Hogan** [1,2], **Michael R. McGowan** [1,*], **Stephen D. Johnston** [2], **Allan T. Lisle** [2] and **Kylie Schooley** [3]

1    School of Veterinary Science, The University of Queensland, Gatton 4343, Australia
2    School of Agriculture and Food Sciences, The University of Queensland, Gatton 4343, Australia
3    Rocky Springs Cattle Co., Mundubbera 4626, Australia
*    Correspondence: m.mcgowan@uq.edu.au; Tel.: +61-7-5460-1856; Fax: +61-7-5460-1922

**Abstract:** Observations of 17 heifer-calf pairs were collected over the first 5 days postpartum (p.p.) to study nursing and maternal behaviour of beef cattle. Cattle were managed in a 6 ha paddock and heifer–calf interactions were determined from both regular visual observations and video recordings. Of 17 potential calves, one was stillborn and 3 died in the first 5 days p.p. associated with dystocia and congenital malformation. Four further potential neonatal calf mortality risks were identified, which included poor calving site selection (n = 1), calf misadventure (n = 1), heifer-calf separation (n = 3) and mismothering (n = 3), with each resulting in distress, physical exhaustion of the calf and impaired nursing. There was marked variation between heifers in the expression of optimal maternal behaviours, with only 39% classified as 'good' mothers. Dam terminated nursing bouts were 27% shorter in duration than calf terminated nursing bouts; 29% of heifers terminated nursing bouts at least 50% of the time. Only 68% of observed suckling interactions were considered effective. On average, the nursing behaviour lasted 7.0 min, with sucking making up 54% of the total nursing time, the sucking rate was 2.0 sucks/s, and the calves performed 7.5 teat-switches, 2.4 butts and 0.9 teat-strips per min of nursing. In 67% of nursing interactions, the calves sucked on all four teats. By three days p.p., all calves developed a clear, consistent suckling pattern. Prior to this, the calves had shorter nursing bouts, spent less time nursing and manipulating the udder, paused more, switched teats and butted less, and had a slower sucking rate. The behaviour of some calves (i.e., low teat fidelity and high levels of milk stimulation behaviours) suggested that their dam milk availability was low. This study has quantified early post-partum nursing behaviour of neonatal beef calves and highlighted dam and calf behaviours that may adversely affect milk intake and, therefore, impact calf survival.

**Keywords:** cattle; maternal behaviour; nursing; primiparous; suckling behaviour



## 1. Introduction

Losses from pregnancy diagnosis to weaning (calf wastage) continue to be an important cause of reduced annual liveweight production in rangeland beef herds, particularly those in tropical environments. McCosker et al. [1] reported that in the extensive rangelands of northern Australia, a minimum of 25% of heifer management groups in commercial beef herds experienced calf wastage of at least 20%. Studies in this region have demonstrated that the period of greatest loss is around the time of calving [2–5]. Perinatal calf losses are considered second only to infertility in contributing to low reproductive performance in beef cattle [6]. Many factors can affect calf survival in the first week after birth; dystocia, weak calf syndrome, maternal death, poor udder and teat conformation, poor mothering ability, congenital abnormalities, neonatal infections, predation and dehydration are all known causes of perinatal loss [7].

Survival of the newborn calf is dependent on the expression of appropriate behaviour, both by its dam and by the calf itself [8]. Important dam components are behaviours that

allow normal dam-offspring bonding to take place, nursing behaviours, responsiveness and attentiveness towards the calf, and protection of the calf [9]. Important calf components are that it must stand and suckle within 1–2 h of birth. The first suckling must occur soon after calving and the lack or delay of it increases the incidence of calf mortality [10]. Successful suckling depends upon the vigour of the calf, its teat-seeking behaviour, the behaviour of the dam, and most significantly, dam udder/teat conformation [5,6,11,12]. Understanding dam-offspring behaviour is critical to identifying situations that increase the risk of calf mortality and management strategies to mitigate this risk.

There have been only a small number of studies of the suckling behaviour of beef calves during the early neonatal period. Previous suckling behaviour studies have focussed on (i) the daily rhythm, frequency and duration of suckling bouts in calves typically ≥7 days old (*Bos taurus* [13–18]; *Bos indicus* [19–23]), or (ii) time to standing, initial teat-seeking behaviour and time to first suckling in calves ≤12 h old, (*Bos taurus* [24]; *Bos indicus* [6]). Lidfors et al. [25] described three phases of behaviour expressed by beef calves during nursing: (i) 'pre-stimulation'—short sucking bouts with a relatively high butting frequency; (ii) 'milk intake'—long, rhythmical sucking bouts with a low butting frequency; and (iii) 'post-stimulation'—short sucking bouts with an initially high butting frequency, which then decreases. Important breed differences in suckling behaviour have been reported in several studies [20,21,23], with differences being attributed to cow milk production and non-nutritive suckling, which has been reported to make up 30–50% of the total nursing bout duration in beef calves [25].

In this study, we examined the behaviour of 17 heifer-calf pairs (from birth to five days p.p.) on a semi-extensive north Australian beef cattle property. Specific aims were to identify the occurrence of optimal/suboptimal maternal behaviour, identify situations that may increase the risk of calf mortality and (iii) describe in detail the behaviours expressed by beef calves during nursing in this environment.

## 2. Materials and Methods

### 2.1. Animals and Study Area

The study was conducted on a commercial beef cattle property in sub-tropical southern Queensland (25°35′ S, 151°18′ E). Seventeen primiparous Angus, Hereford and Shorthorn cross heifers (approximately 2 years of age at time of calving), which were conceived for fixed-time artificial insemination, were selected for the study. To facilitate the planned intensive behavioural observations, the heifers were induced to calve with a single injection of 500 ug cloprostenol (prostaglandin F2$\alpha$) 280 days after AI. They were then placed in a 6.1 ha paddock (Figure 1) adjacent to the cattle handling facility. The heifers grazed mixed sub-tropical pasture consisting of Rhodes (*Chloris gayana*), buffel (*Cenchrus ciliaris*) grasses and tropical legumes (*Aztec siratro*, *Seca stylo* and *Desmathus* spp.). They also had ad lib access to water from a trough and were fed a fortified molasses supplement containing 8% cottonseed meal, 3% urea and 1% dicalcium phosphate in a trough (2.5 to 3.5 kg/d/animal). Calving took place in September (late dry season). No rainfall was recorded during the study period and the average maximum and minimum temperatures were 27 °C and 8 °C, respectively. Following induction of parturition, the heifers were monitored for signs of parturition at least twice daily [26]. Ease of calving was scored on a four-point scale: (i) easy, unassisted; (ii) easy, assisted; (iii) difficult, assisted; (iv) difficult, requiring veterinary assistance [26], and the calving site of each calf was recorded (Figure 1). The study was approved by The University of Queensland's Animal Ethics Committee (SVS/346/12).

### 2.2. Monitoring of Cattle Behaviour

Behaviour was monitored using a combination of visual observations and video recording of nursing events, focussing in particular on calf suckling. All heifer-calf associations and calf nursing behaviours were recorded as hand-written notes by a single highly experienced observer (LH), who monitored the animals on foot at a distance of 3–10 m. The heifers were exposed to the presence of the observer from the time of induction

of parturition and they appeared to become accustomed to the presence of the observer quickly. All heifers had been managed using 'low stress' handling procedures at weaning and subsequently.

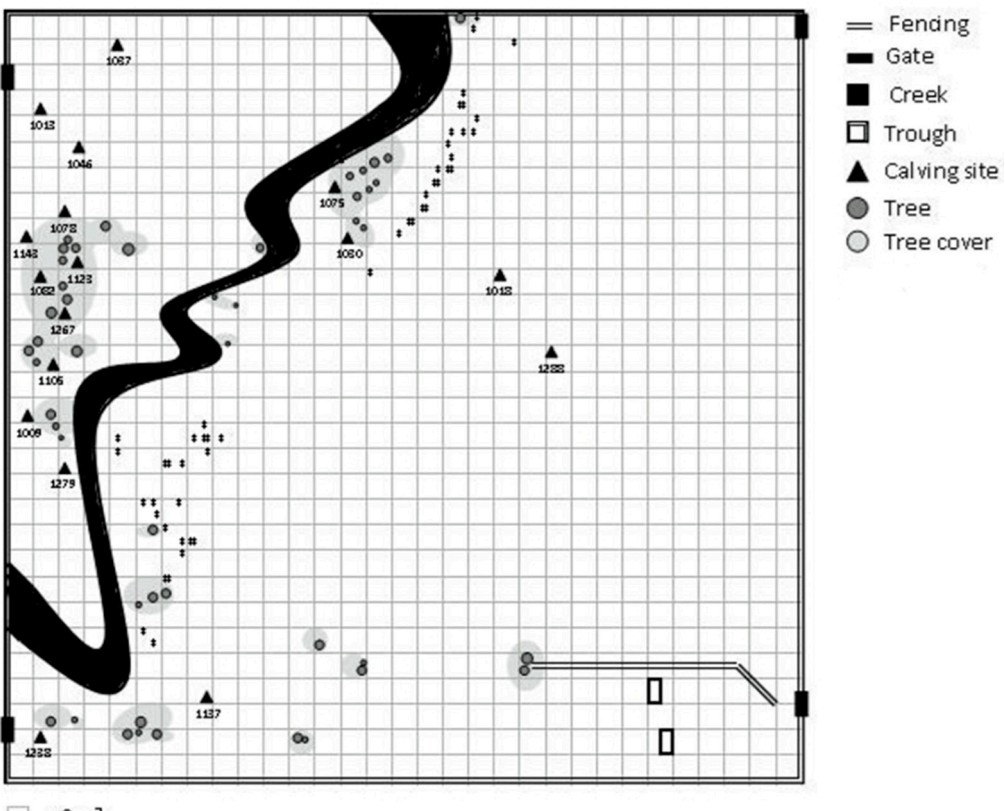

**Figure 1.** Overhead schematic view of the 6.1-hectare calving paddock, including calving sites with the cow number that calved at each site (‡ native shrubbery).

During the observation periods (06:00–13:00 h and 15:00–18:00 h [10 h/day], for five consecutive days p.p. per heifer-calf pair), the observer continuously scanned the group of animals and (i) recorded the nursing behaviours described in Table 1, (ii) recorded the occurrence of the maternal behaviours described in Tables 2 and 3, (iii) identified potential calf mortality risks (situations that resulted in either physical exhaustion of the calf or missed nursing opportunities) and (iv) filmed (fragments, not complete recordings) as many nursing events as possible (60 s minimum to 300 s maximum). Nursing behaviour was filmed using a portable digital video camera (Panasonic Model No. PV-GS400, Panasonic Asia Pacific Pte Ltd., Singapore) at a distance of 2–5 m. Using MotionDV Studio software, Version 5.1LE, Panasonic Asia Pacific Pte Ltd., Singapore, tape recordings were saved as audio video interleave (AVI) files onto a computer and analysed for the duration and frequency of all observable calf suckling behaviours (Table 1). Animals were not observed between 13:00–15:00 h each day as previous studies reported that the frequency of nursing events were reduced during this period of the day [16,18], but also because this was the warmest part of the day when cattle would typically seek shade and rest. Due to the number of heifers calving over a relatively short period, and the terrain of the calving paddock, it was not possible to determine the interval from calving to first suckling, or to observe in all heifers, the immediate behaviour after birth.

**Table 1.** Nursing behaviours were recorded during each observation period.

| Behaviour | Definition of Behaviour | Reference |
|---|---|---|
| Suckle attempt | When a calf tried to get a teat into its mouth without success. | [23] |
| Successful suckling | When a calf succeeded in getting a teat into its mouth, accompanied by observable sucking and swallowing. | [23] |
| Nursing bout | Nursing behaviours had to be expressed for $\geq$60 s to be classified as a nursing bout and a break period (i.e., period of no nursing behaviour) of $\geq$60 s had to occur for bouts to be considered as two separate events. Total duration, bout frequency and bout duration were measured. | [19,27] |
| Bout termination | The heifer was the terminator of a nursing bout when she showed clear activity leading to the interruption of that bout (i.e., walking away, threat or aggression), while the calf was the terminator when no such heifer activity was observed and the calf distanced its muzzle from the udder. | [25] |
| Sucking | The teat was in the calf's mouth as it performed sucking movements for more than 3.5 s. Duration in s/min was recorded. | [25,28] |
| Manipulating | The calf's muzzle was in contact with a teat and it sucked or had a teat in its mouth for less than 3.5 s at a time, as well as butted or stripped the teat. Duration in s/min was recorded. | [28] |
| Pause | The calf's muzzle was off the teat for more than 3.5 s. Duration in s/min was recorded. | [28] |
| Butting | Prodding or striking of the udder by the calf with its muzzle. Frequency/min was recorded. | [28] |
| Teat-change | Recorded whenever the calf left the teat that it was sucking moved to another teat and continued sucking. The teat suckled was identified each time the calf changed teat so that teat fidelity could be calculated. Frequency/min was recorded. | [29] |
| Teat-stripping | The calf pulled the teat downwards or sideways before releasing it. Frequency/min was recorded. | [28] |
| Sucking rate | The number of sucking movements/s was calculated by using the time the calf took to do 20–40 consecutive sucking movements. During each nursing, 1–3 measures of sucking rate were taken, with measures being separated by at least 60 s of nursing. | [28] |
| Nursing position | Three body positions of the nursing calf were distinguished; left-hand side (LHS) and right-hand-side (RHS) nursing, where the calf stood parallel (0–45°) to the dam's body, as well as suckling from behind between the dam's hind legs (REAR) | [30] |

**Table 2.** Summary of suboptimal maternal behaviour identified in cattle.

| Behaviour Unit | Description of Behaviour | Reference |
|---|---|---|
| Parturition | Standing when calving (the risk of calf death is higher when the dam delivers in a standing rather than recumbent position). | [23] |
| | Latency (>5 min) to stand following calving (dams should be standing and initiate licking of their calves almost immediately after calving). | [31,32] |
| | Selection of an unsuitable calving place. | Seen in the current study |
| Bonding | Latency (>5 min) or refusal to lick calf following calving. | [26] |
| | Lack of maternal instinct: difficulty or failure to locate calf after planting it. | [4] |
| Nursing | Delay of calf's first suckling: butting and kicking the calf during initial teat-seeking advances (results from fear of calf and high udder sensitivity). | [11,12,31] |
| | Delay of calf's first suckling: over attentiveness towards the calf, i.e., constant licking, touching and pushing of the calf while it's making teat-seeking advances. | [6] |
| | Standing after calving is short: dam lies down again before its calf has successfully suckled or at least made numerous suckling attempts. | [11,31] |
| | Low suckling rate: a low frequency of acceptance of the calf's suckling attempts (may result in total rejection of calf). | [31,33,34] |
| Responsiveness | Dam down (frequency dam observed lying while its calf is active >5%). | [11,33] |
| | Low interest: dam doesn't pay attention/is not interested in its calf when it struggles to rise or makes teat-seeking advances. | [31] |
| | Following prolonged separation (>3 h), the dam does not perform reuniting behaviours, e.g., increases in vocalizations, licking/muzzling of the calf and increased activity. | [35] |

**Table 3.** Summary of optimal maternal behaviour identified in cattle.

| Behaviour Unit | Description of Behaviour | Reference |
|---|---|---|
| Parturition | Recumbent when calving.<br>Dam rises immediately (<5 min) after calving and initiates licking of the calf.<br>Selection of a suitable calving place. | [23]<br>[31–33]<br>[36] |
| Bonding | Licking of the calf immediately (<5 min) after calving (important in establishing a strong bond between dam and calf).<br>High maternal instinct (ease in locating calf after 'planting' it). | [26]<br>[4] |
| Nursing | Acceptance of initial calf teat-seeking advances (dam remains stationary or makes postural changes that make teat-seeking/suckling easier for the calf).<br>Standing after the calf is delivered (dam does not lie down again until its calf has successfully suckled or made numerous suckling attempts).<br>High suckling rate (little to no rejection of calf suckling advances, often accompanied by licking of the calf). | [11,12,31,33]<br><br>[11,31]<br><br>[31,34] |
| Responsiveness | Dam down (frequency dam observed lying while its calf is active <5%).<br>High interest (dam pays attention/is interested in its calf when it struggles to rise or makes teat-seeking advances).<br>Following prolonged separation (>3 h) dam increases its vocalizations and activity, which serve to reunite the dam and calf. | [11]<br>[31]<br>[35] |

*2.3. Data Analysis*

The duration and frequency of each nursing behaviour were summarized for each calf by day after calving. The mean total nursing time, the number of nursing bouts and bout length were calculated from an aggregation of data from all calves (n = 14) overall sampling days related to that period after calving (0–24 h = 1 day p.p., 24–48 h = 2 days p.p., 48–72 h = 3 days p.p., 72–96 h = 4 days p.p. and 96–120 h = 5 days p.p.). Patterns of nursing activity were determined by dividing the data into $10 \times 1$-h time increments, starting at 06:00 h and ending at 18:00 h, with, for example, all nursing behaviour recorded between 06:00–06:59 being grouped into the 06:00 h time period. This data was then analysed to determine the overall total mean time spent nursing per calf per hour, with total time being expressed as sucking per hour (s/h).

Residual data sets were tested for normal distribution and when the original scale violated the homoscedasticity assumption, a logarithmic transformation ($\log_{10}$) was used to achieve a normal distribution for statistical analysis. The calculation of statistical tests was carried out using the programs Minitab (Version 15.1, 2007) and SAS (SAS®/STAT, Version 9.2, 2010, State College, PA, USA), with all significance levels set at $p \leq 0.05$. Results from the statistical analysis are reported as least square means (LSM) with standard error (SE) unless otherwise noted. A mixed-model ANOVA with REML estimation was used to test age and terminator effects on the total duration of nursing, the number of nursing bouts, and bout length. Between (calf) and within (residual) subject effects were random, while age and terminator effects were fixed. The calf age (number of days after birth) effect was calculated overall (days 1–5 after calving) and by partitioning the age effect into two independent components (1 d vs. 2–5 d; other ages: 2–5 d). A FREQ procedure was performed to produce a one-way frequency table for the numbers and percentages of dam and calf terminated nursing bouts.

A mixed-model ANOVA with REML estimation and a weighed variable was used to test calf age effects on the duration and frequency of individual nursing behaviours. Between (calf) and within (residual) effects were random, while age effects were fixed. Sucking rate (sucks/s) was determined as follows the number of sucking rates measured/nursing, while teat-switching (number/min), butting (number/min), teat-stripping (number/min), sucking (s/min), manipulating (s/min) and pause (s/min) was measured against video duration (i.e., number of seconds of nursing captured on film/nursing). The calf age effect was calculated overall and by partitioning the age effect into two independent components (1 d vs. 2–5 d; other ages: 2–5 d).

To rank the heifers according to the quality of their observed maternal behaviour (best to worst), an odds ratio [odds (optimal)/odds (suboptimal)] of each heifer expressing optimal rather than suboptimal maternal behaviour was calculated (Tables 2 and 3). Odds were calculated as [(number of optimal behaviours observed + 0.5)/(potential number of optimal behaviours that could have been observed—the actual number of optimal behaviours observed + 0.5)] and [(number of suboptimal behaviours observed + 0.5)/(potential number of suboptimal behaviours that could have been observed—the actual number of suboptimal behaviours observed + 0.5)].

## 3. Results

*3.1. Calving Site and Outcomes*

While most heifers calved 24–72 h following the prostaglandin F2α injection, one heifer failed to calve and was removed from the study. Calving sites were not randomly distributed throughout the available area (Figure 1). Of the heifers studied, seven separated from the group at calving (Figure 1). Most calved on dry, higher elevations of the paddock with either tall grass or tree cover. Two heifers (1030 and 1075) calved on damp lowland, 1–2 m away from the edge of the creek with tree cover (Figure 1). One heifer (1279) calved close (~1 m) to a steep bank into the creek.

Eleven heifers calved unassisted, while two required minor assistance; heifer 1018 required gentle manual extraction of the foetus, and heifer 1267 required manual assistance

to stand the following calving. Four heifers (1009, 1105, 1143 and 1238) had a 'difficult vet assisted' calving. All four had dystocia due to foetal-maternal disproportion; 3/4 of calves were successfully delivered alive by forced extraction. Heifers 1009 and 1105 both subsequently developed signs of metritis and were treated with an intramuscular antibiotic by the attending veterinarian; 1105 made a full recovery and started nursing her calf within 24 h p.p., while 1009 did not respond to treatment and was euthanised four days p.p.

Of 17 potential live-born calves, one was still-born, one died during birth as a consequence of severe dystocia, one that did not require assistance to be delivered was observed unable to stand or nurse following birth and died within 24 h, and one died 5 days after birth due to difficulties suckling associated with a cleft lip.

### 3.2. Neonatal Calf Mortality Risks

Four potential risks were identified and included poor calving site selection, excessive moving of the newborn calf, heifer-calf separation and calf mismothering. Heifer 1279 calved close to a steep bank extending into the creek, and shortly after birth, her calf fell down this bank and landed in the creek. The calf would have perished if not for human intervention, as it appeared to become exhausted quickly and was unable to get out of the creek. During the first 24 h p.p. heifer 1078 crossed the creek twice with her calf. During the second crossing, her calf became stuck in the creek bank mud, and after 1 h of struggling to free itself, human intervention was again required to rescue it.

Due to their size, calves could pass through the paddock's barbed wire fencing and enter the adjacent paddock; this was observed in three heifer-calf pairs. Heifer 1078 and her calf both became distressed in response to this physical separation, defined by a major increase in vocalization and activity; within 1 h, this calf managed to pass back through the fence and reunite with its dam. Heifer 1267 and her calf also became physically separated and appeared similarly distressed. However, despite repeated attempts, the calf was unable to pass back through the fence and then subsequently laid down. After 5 h of separation, human intervention reunited the calf with the dam; consequently, this calf had no nursing opportunities 18–22 h p.p. By contrast, neither heifer 1046 nor her calf showed any evidence of separation distress, despite the calf being on the wrong side of the fence for over 7 h. While human intervention eventually reunited the calf with her dam, the calf had no nursing opportunities for approximately 21–30 h p.p.

Varying degrees of mismothering were observed in 3 calves; these calves were regularly observed to 'wander and bellow out' to their dams (heifers 1037, 1009 and 1046), whereas the other calves were typically silent when unaccompanied by their dam. Heifer 1037 had a low maternal suckling rate (i.e., a low frequency of acceptance of her calf's suckling attempts) and her calf regularly resorted to suckling from another dam (heifer 1143). Heifer 1009 had low milk production following dystocia and her calf had to be supplemented with 2–3 hand-feeds per day for the first three days p.p. Heifer 1046 frequently abandoned her calf for extended (≥4 h) periods of time and paid little attention to the calf during nursing.

### 3.3. Heifer Rankings

Heifers were ranked in terms of maternal behaviour expressed (best-to-worst) 1013 and 1078, 1030, 1137, 1267, 1082, 1075, 1238, 1279, 1046, 1123, 1105, 1037 and 1009 (Figure 2). In terms of the percentage of nursing bouts terminated by the dam, the heifers ranked (best-to-worst) 1030 (8.3%), 1075 (11.1%), 1105 (25.0%), 1082 (27.3%), 1137 (28.6%), 1279 (31.6%), 1037 (38.5%), 1238 (43.8%), 1013 (48.2%), 1078 and 1267 (50.0%), 1009 (66.7%), 1046 (71.5%) and 1123 (84.6%). Comparing both rankings, 2 heifers were ranked in the 'best' 5 (1030, 1137) and 'worst' 5 (1123, 1009), respectively, for both maternal behaviour and percentage dam terminated nursings.

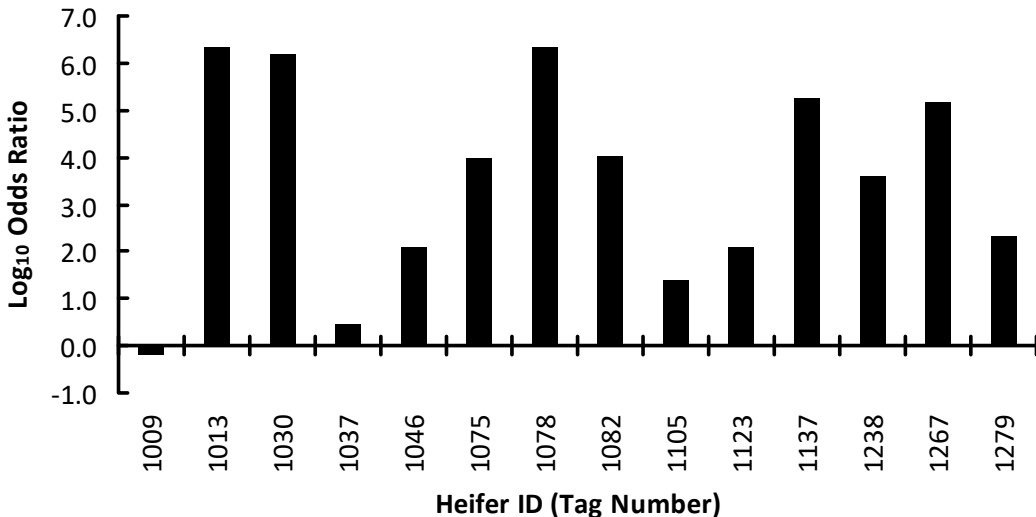

**Figure 2.** Odds ratio (OR) of each heifer expressing optimal v's suboptimal maternal behaviour.

*3.4. Nursing Activity: Timing, Duration and Frequency*

A total of 206 nursing bouts and 98 suckle attempts were recorded. Failed suckling occurred mainly due to rejection by the dam (65.3%) or when the calves stopped their suckle attempt (29.6%) or were rejected by another dam during an allosuckling attempt (5.1%). Sunrise (during this study) ranged from 05:53–06:06 h and the first peak in nursing activity was observed shortly after this (06:00–07:00 h; Figure 3A). The second peak in nursing activity occurred from 12:00–13:00 h, while the most regular nursing time occurred between 16:00–18:00 h, just prior to sunset (range: 17:43–17:48 h) (Figure 3A). Nadirs in nursing activity were observed at 09:00–10:00 h and 15:00–16:00 h when most heifers were observed grazing or consuming the molasses-based supplement.

The total mean nursing bout duration was $7.0 \pm 0.3$ min (range: 1.0–21.9 min), with 80% of the nursings lasting 3–9 min (Figure 3B). Nursing bout duration was affected by calf age ($F_{4,41} = 3.45$, $p = 0.02$) and dam termination ($F_{1,146} = 11.90$, $p < 0.01$). Calves 1 d old had a shorter mean nursing bout duration ($-2.1 \pm 0.6$ min) than 2–5 d calves ($F_{1,41} = 10.58$, $p < 0.01$), but there was no difference in mean nursing bout duration amongst 2–5 d calves ($F_{3,41} = 1.25$, $p = 0.31$; Figure 4A). When the dam terminated nursing, the bout was shorter ($-1.9 \pm 0.6$ min) than when the bout was calf terminated. Whether a nursing was terminated by the dam or calf was a function of the dam-calf pair ($X^2 = 34.54$, $p < 0.01$, DF = 13), not calf age, ($X^2 = 2.52$, $p = 0.65$, DF = 4), as there were no significant calf age effects overall (1–5 d: $F_{4,41} = 1.25$, $p = 0.31$), 1 d vs. 2–5 d ($F_{1,41} = 1.02$, $p = 0.36$) or 2–5 d ($F_{3,41} = 1.28$, $p = 0.30$) (Figure 4B).

The total mean daily time (min/10 h) spent nursing was $25.5 \pm 2.7$ min, with a range from 17.2 to 38.1 min. The overall effect of calf age on mean nursing time was marginal ($F_{4,41} = 2.18$, $p = 0.09$), but there was an effect of the calf being 1 d old. Calves 1 d old had a shorter mean total daily nursing time ($-8.4 \pm 4.0$ min) than 2–5 d calves ($F_{1,41} = 4.48$, $p = 0.04$; Figure 4A). There was no difference in total mean daily nursing time amongst 2–5 d calves ($F_{3,41} = 1.71$, $p = 0.18$). The total mean daily nursing frequency (number/10 h) was $3.5 \pm 0.3$, with a range from 1 to 9. Overall effect of calf age on nursing frequency was marginal ($F_{4,41} = 2.37$, $p = 0.07$) and there was no effect of the calf being 1 d old ($F_{1,41} = 0.01$, $p = 0.90$). There was a marked difference in total mean daily nursing time amongst 2–5 d calves ($F_{3,41} = 3.14$, $p = 0.03$), but there was no obvious calf age-related pattern (Figure 4A).

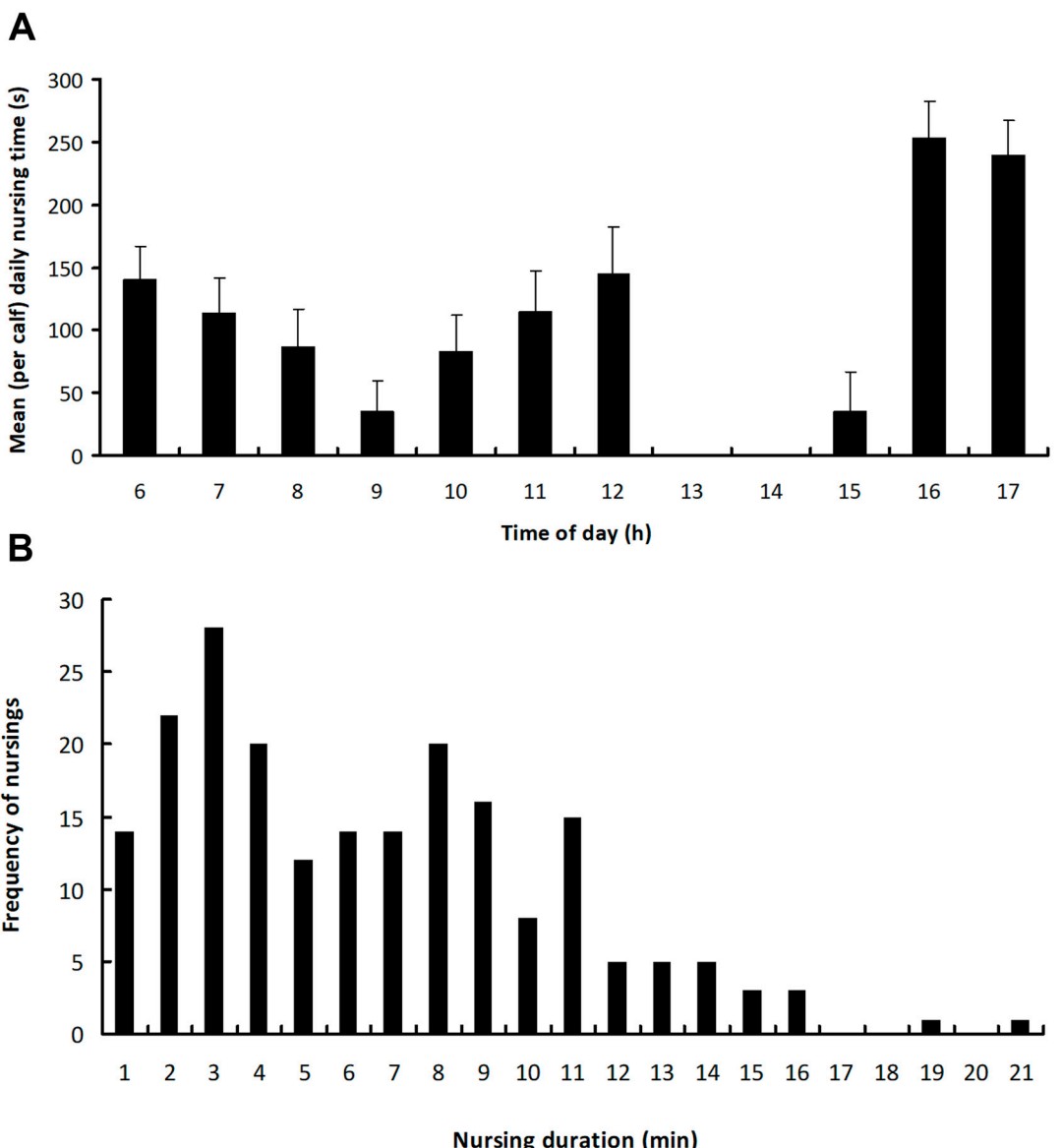

**Figure 3.** (**A**) Mean (±SE) nursing duration (s/h per calf) of beef calves (n = 14) during the first five days postpartum; (**B**) Frequency distribution according to nursing duration (min) in beef calves during the first five days postpartum. Columns for each hour (e.g., 6) represent activity in the following hour (i.e., 06:00–06:59).

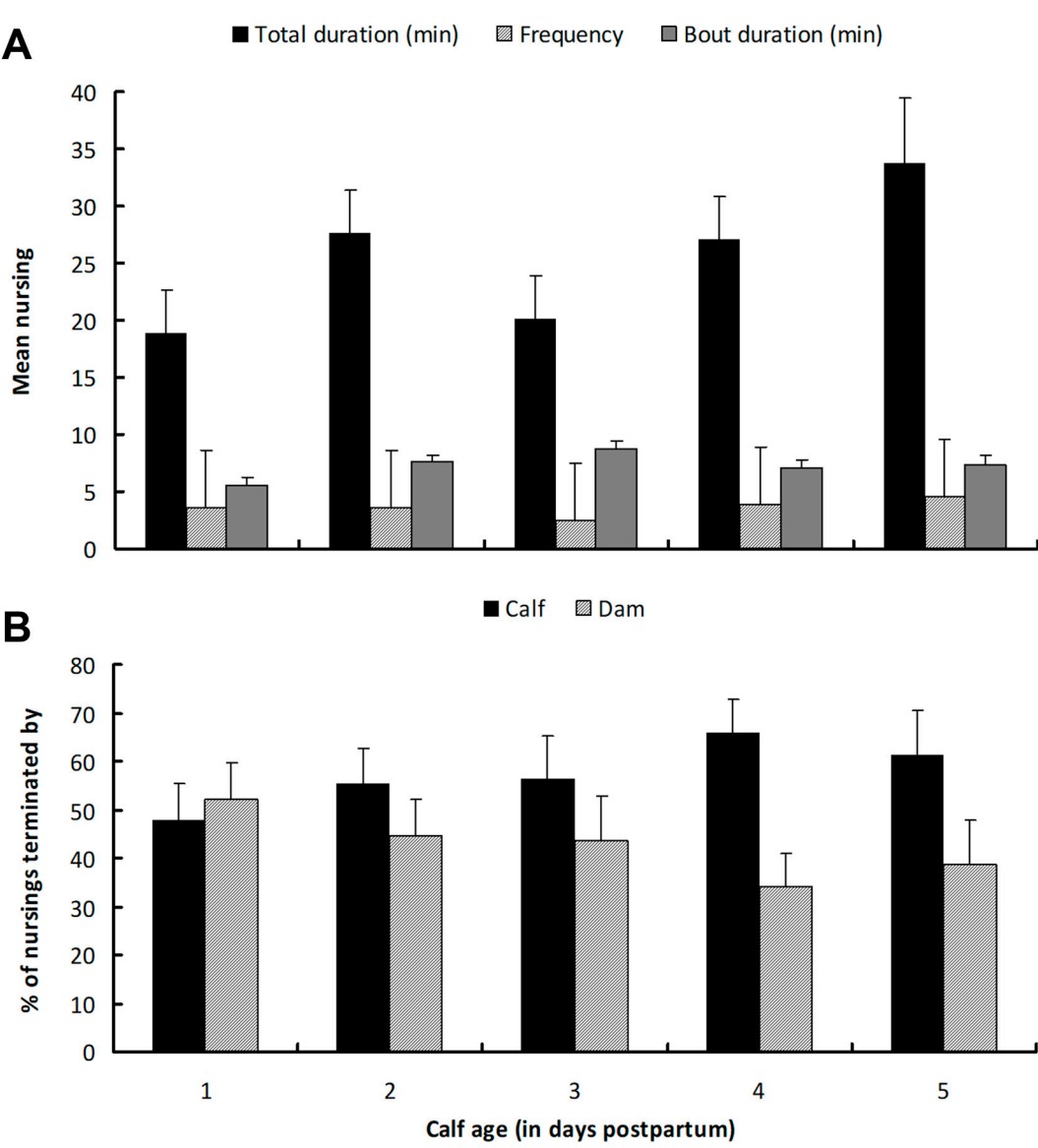

**Figure 4.** (**A**) Mean (± SE) daily total nursing duration (min), nursing frequency and nursing bout duration (min) of calves 1–5 d; (**B**) Percentage of nursings that were terminated by dam or calf at 1–5 d. 1 d, n = 46; 2 d, n = 47; 3 d, n = 32; 4 d, n = 50; 5 d, n = 31.

*3.5. Nursing Behaviours (Frequencies and Preferences)*

Total (n = 125) mean sucking rate (sucks/s) was 2.01 ± 0.03, with a range from 1.08 to 2.67. Overall effect of calf age on sucking rate was significant ($F_{4,42} = 13.40$, $p < 0.01$) with: (i) 1 d calves having a lower mean sucking rate ($-1.70 ± 0.06$) than that of 2–5 d calves ($F_{1,42} = 38.47$, $p < 0.01$; Figure 5A); and (ii) 2 d calves having a lower mean sucking rate ($-1.87 ± 0.07$) than 3–5 d calves, but 3–5 d calves having comparable mean sucking rates ($F_{3,42} = 5.50$, $p < 0.01$; Figure 5A). Sucking made up 53.6% of the total nursing duration, manipulating the udder accounted for 33.0% and paused accounted for 13.4%. The amount of sucking (s/min) during nursing was similar whatever the age of the calves (1–5 d: $F_{4,52} = 2.43$, $p = 0.06$; 1 d vs. 2–5 d: $F_{1,52} = 3.18$, $p = 0.08$; 2–5 d: $F_{3,52} = 1.95$, $p = 0.13$), but the amount of manipulating ($F_{4,52} = 4.30$, $p < 0.01$) and pause ($F_{4,52} = 12,37$, $p < 0.01$) during nursing (s/min) varied according to calf age (Figure 5B). For manipulating: (i) 1 d calves manipulated the same as 2 d calves, but less ($-7.1 ± 3.5$ s/min) than 3–5 d calves ($F_{1,52} = 6.29$, $p = 0.02$); and (ii) 2 d calves manipulated less than 3–5 d calves, with 3–5 d calves manipulating the same ($F_{3,52} = 3.84$, $p = 0.01$). For pause: (i) 1 d calves paused more ($+12.4 ± 2.3$ s/min) than 2–5 d calves ($F_{1,52} = 35.37$, $p < 0.01$); and (ii) 2 d calves paused

less than 1 d calves, but more (+5.87 ± 2.7 s/min) than 3–5 d calves who paused the same ($F_{3,52} = 3.74$, $p = 0.02$).

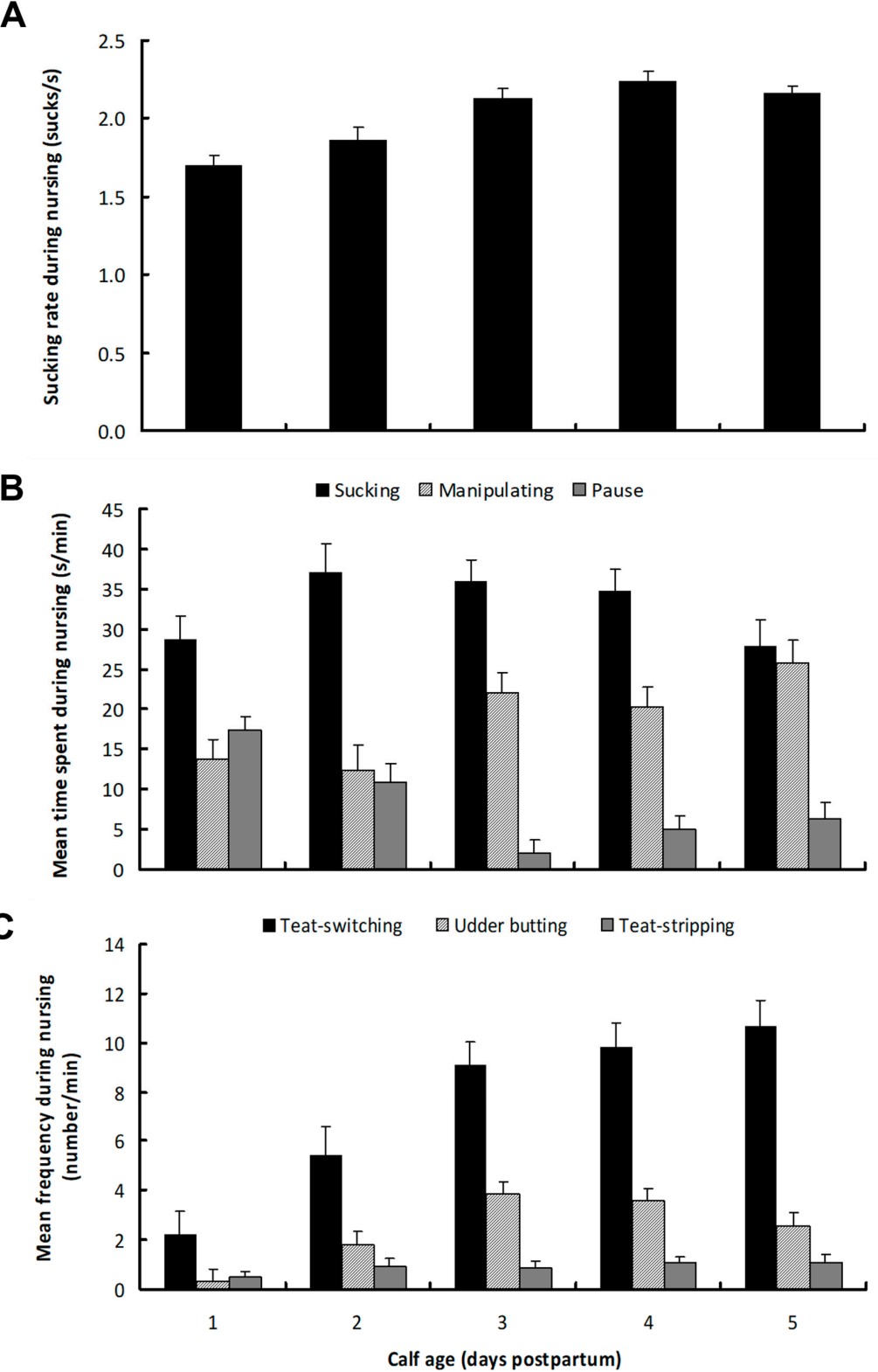

**Figure 5.** Mean (± SE) (**A**) sucking rate (sucks/s); (**B**) time spent sucking, manipulating or at pause (s/min); and (**C**) frequency of teat-switching, udder butting and teat-stripping (number/min) during nursing for 1–5 d beef calves (n = 14). 1 d, n = 16; 2 d, n = 10; 3 d, n = 14; 4 d, n = 17; 5 d, n = 13.

On average, the calves performed $7.5 \pm 1.0$ teat-switches, $2.4 \pm 0.5$ udder butts and $0.9 \pm 0.3$ teat-strips (Table 1) per minute of nursing (Figure 5C). The number of teat-strips (number/min) during nursing was similar whatever the ages of the calves (1–5 d: $F_{4,52} = 1.03$, $p = 0.40$; 1 d vs. 2–5 d: $F_{1,52} = 3.71$, $p = 0.06$; 2–5 d: $F_{3,52} = 0.16$, $p = 0.92$), but the number of teat-switches ($F_{4,52} = 19.61$, $p < 0.01$) and udder butts ($F_{4,52} = 13.61$, $p < 0.01$) during nursing (number/min) varied according to calf age (Figure 5C). For teat-switching: (i) 1 d calves switched teats less often ($-6.8 \pm 1.0$/min) than 2–5 d calves ($F_{1,52} = 66.49$, $p < 0.01$); and (ii) 2 d calves switched teats more often than 1 d calves, but less often ($-4.3 \pm 1.2$/min) than 3–5 d calves who switched teats the same ($F_{3,52} = 5.59$, $p < 0.01$). For udder butting, (i) 1 d calves butted less often ($-2.2 \pm 0.6$/min) than 2–5 d calves ($F_{1,52} = 36.70$, $p < 0.01$) and (ii) 2 d calves butted more often than 1 d calves, but less often ($-1.5 \pm 0.6$/min) than 3–5 d calves who butted at the same rate ($F_{3,52} = 5.27$, $p < 0.01$).

One, two, three, or four teats were sucked in 3.0%, 13.4%, 16.4% and 67.2%, respectively, of the observed nursings. The front teats were sucked 63.5% and the rear teats 36.5% and there were no clear changes associated with age (Figure 6A). Calves nursed from the RHS, LHS and REAR of the dam 55.5%, 40.9% and 3.6% of the time, respectively. One and 2 d calves favoured the LHS of the dam, while 3–5 d calves favoured the RHS of the dam (Figure 6B). REAR nursing was infrequent and was mostly associated with heifer 1037, who rarely accepted her calf's suckling attempts.

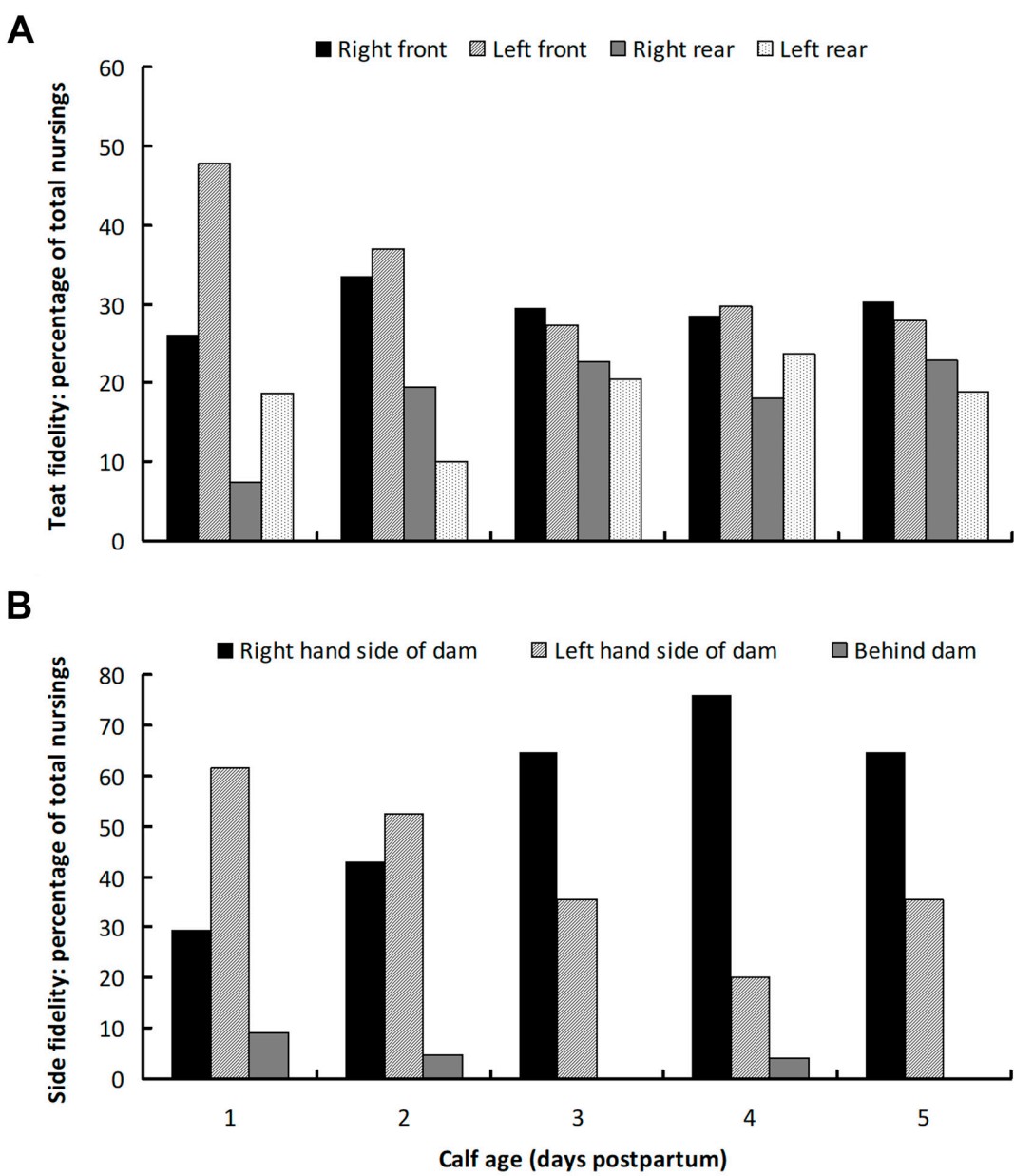

**Figure 6.** Percentage of total nursing time spent (**A**) on the right front, left front, right rear or left rear teat; and (**B**) suckling from the right-hand side, left-hand side, or rear (i.e., behind) of the dam, in 1–5 d beef calves (n = 14). 1 d, n = 16; 2 d, n = 10; 3 d, n = 14; 4 d, n = 17; 5 d, n = 13.

## 4. Discussion

This study presents quantitative data in an attempt to help define the early postpartum maternal behaviour of beef heifers and the suckling behaviour of their calves. Although the frequency of dystocia was higher than expected for this herd, it was within the reported range for heifers calving at approximately 2-years of age [37]. Further, although an increased prevalence of retained foetal membranes is a common sequela of induction of parturition, only 2 heifers in this study required post-partum antibiotic treatment.

Similar to other studies [11,17,36], the current study found that only some cattle (7/17) seek isolation at calving, with a suitable calving site apparently being more important to the dam than isolation from the herd. Similar to the findings of Lidfors et al. [36], most heifers in this study sort higher ground and dry sites with tall grass or tree cover to calve. This behaviour is likely to be important as studies have found that air temperature and

rainfall affect neonatal calf survival, with greater mortality seen during cold and wet weather [6,38,39].

In extensive beef cattle production systems, the responsibility of rearing the newborn calf is entirely the heifer/cow's; it is, therefore, essential that the dam be able to successfully rear her offspring without human intervention [9]. In this study, where a small group of grazing heifers were regularly monitored prior to and following parturition, 4 heifers had to be treated for dystocia, with one calf dying as a consequence, 2 calves became stuck in a creek and 2 calves became separated from their dams, all of which required human intervention. In a further two cases, 1 heifer required gentle traction to deliver its calf and the other experienced mild obstetrical paralysis, both potentially causing a delay in first suckling. Nevertheless, despite all the live-born calves from these heifers experiencing varying degrees of distress, physical exhaustion and delays in suckling, all survived and were subsequently weaned. The relatively benign weather conditions, easy access to water, grazing and supplement are likely to have contributed to this outcome. Further, it is acknowledged that placing the 17 heifers in a small paddock to facilitate observations may have contributed to some of the observed events (calf separation from the dam).

As the neonatal calf typically has low energy stores, it needs to obtain an adequate volume of milk at intervals of no longer than 6–7 h [40]. The consequences of inadequate milk intake are reduced growth and increased risk of disease and mortality. In the hot conditions, which typically occur during the calving period in the tropical rangelands of northern Australia, Fordyce et al. [41] have demonstrated that neonatal calves with no access to milk or water can become 20% dehydrated in 1–3 days, thereby putting them at serious risk of mortality.

An important component of maternal success is 'good' maternal behaviour. As defined in Tables 2 and 3, this consists of a range of behaviours from a selection of a suitable calving site to acceptance of initial calf teat seeking and continuing to show little to no rejection of calf suckling advances. The emphasis in recent years on the improvement of mothering ability through selection programs has increased the importance of information concerning factors associated with mothering ability [9]. The purpose of our 'heifer rankings' in the current study was to explore behaviour traits as potential selection traits for improved maternal behaviour and calf survivability. The heifers showed wide variability in their ratios of optimal-to-suboptimal maternal behavioural expression. Heifers 1013, 1078, 1030, 1137 and 1082 expressed a high incidence ($\geq$5.5:1) of optimal-to-suboptimal behaviour and were classified as 'good' mothers. Heifers 1075, 1238, 1279, 1046 and 1123 expressed a moderate incidence (2–4:1) of optimal-to-suboptimal behaviour and were classified 'average' mothers, while heifers 1105, 1037 and 1009 expressed a low incidence ($\leq$1.5:1) of optimal-to-suboptimal behaviour and were classified 'poor' mothers. Although the heritability of maternal behaviour in beef cattle has been reported to be low, candidate genes have been identified [42], which may facilitate selection for improved maternal behaviour.

Empirical work has shown that hungry calves bellow more and spend more time standing than calves that are sufficiently fed [29,43]. Cattle are described as hiders in terms of the mother-young relationship after parturition, and calves are normally silent during the first few days p.p.; as a strategy to avoid predation [43]. In this study, mismothering by heifers 1046, 1037 and 1009 resulted in their calves frequently 'wandering and bellowing,' which may increase the risk of predation and misadventure.

An important finding of the present study was further confirmation that dam termination of nursing bouts is an undesirable maternal trait, as dam terminated nursing bouts were, on average, 27.1% shorter in duration than calf terminated nursing bouts. In this study, the percentage of dam terminated nursing bouts ranged from 8.3–84.6% (per individual dam) and were typically higher than previously reported beef heifer termination rates (58.8%, [11]; 18.2–25.0%, [25]. It is not clear to what extent the dam actively terminated nursing bouts and if this changes with age [35], but nursing bouts should mostly be terminated by the calf [25].

Of the suckling interactions observed in this study, only 68% were considered effective, which is much lower than that reported by Paranhos de Costa et al. [23] for Zebu and Criollo calves. This is likely due to the fact that the study conducted in Brazil involved cows, whereas the dams in the present study were heifers. Differences may also be due to calf age and breed differences (See [14,20,21]. Further, greater udder sensitivity in heifers following parturition and new-born calf uncertainty during early teat-seeking advances may have also contributed to the high proportion of dam terminated suckling observed in this study. Similar to previous studies, the highest rates of suckling took place in the early morning and in the late afternoon, with most calves suckling 3–5 times per day [13,16,18].

A previous study on the temporal patterning of suckling bouts in cross-bred beef cattle indicated that calves develop a clear and effective suckling pattern somewhere between 24 h and 7 day p.p. [25]. The present study managed to narrow this window to between 2–3 day p.p., with all calves showing a consistent suckling pattern from 3 d. Calves 1–2 d old had shorter nursing bout durations, spent less time nursing, had a slower sucking rate, spent less time manipulating the udder, paused more, switched teats less, performed less butting and favoured nursing from the left-hand-side (LHS) of the dam. From 3 day p.p., these suckling traits stabilized and were similar in value to those found in previous studies for calves ranging 1–180 d in age (Table 4). Demonstrating that the neonatal beef calf takes several days after birth to establish a nursing pattern further confirms how critical this period is with respect to calf survival. Muller et al. [44] recently reported that about a third of *Bos indicus* cross beef calves might initially experience sub-optimal growth due to inadequate milk intake. Using daily weight gain of calves as a proxy for milk intake, they identified two different neonatal growth profiles; calves which gained at least 0.5 kg/day from birth and calves, which often gained only 0.2 kg/day and did not achieve adequate growth, until day 3 after birth.

**Table 4.** Summary of reported suckling behaviour of beef and dairy calves.

| Reference | Total min/d | No./d | Bout (min) Duration | Sucks/s | Teat-Switch | Butts (No./min) | Teat-Strips | Sucking (s/min) | Manipulate (s/min) | Pausing (s/min) |
|---|---|---|---|---|---|---|---|---|---|---|
| Current study Beef: 3–5 d (10 h/d) | 30.0 ± 4.2 | 3.6 ± 0.6 | 7.7 ± 0.7 | 2.2 ± 0.1 | 9.9 ± 1.0 (/min) | 3.3 ± 0.5 | 1.0 ± 0.3 (/min) | 32.9 ± 2.9 (54.8%) | 22.7 ± 2.6 (37.8%) | 4.4 ± 1.8 (7.4%) |
| [14] Beef: 1–120 d (12 h/d) | 30–35 | 3.0–3.5 | 10–11 | | | | | | | |
| [25] Beef: 1–123 d | | | 4.0–32.1 | | | | | 18.5–35.2% | 39.3–50.0% | 10–25% |
| [20] Beef: 1–180 d (12 h/d) | 18.3–20.6 | 1.6–1.9 | 7.0–8.2 | | | | | | | |
| [45] Beef: 1–4 d (24 h/d) | 51 (43.1–61.3) | 6.9 (5.7–8.0) | 7.6 (6.4–8.8) | | | | | | | |
| [23] Dairy: 30–120 d (12 h/d) | 23.7 ± 0.5 | 2.6 ± 0.1 | 9.3 ± 0.1 | | | | | | | |
| [29] Dairy: 7–49 d | | | 9.8–11.8 | 2.2 ± 0.2 | 25.0–37.8 (/nursing) | 0.7 (high) 2.5 (low) | | 40–80% | | |
| [28] Dairy: 7,14 & 28 d | | | 7.2 (2.8–16.3) | 2.1 ± 0.02 | 0.41 ± 0.13 (/min) | 1.5 (1.0–2.4) | 0.02 ± 0.01 (/nursing) | 66.4% | 9.8% | 22.2% |

The sucking rate appears constant throughout nursing, and the average rate (sucks/s) is similar irrespective of nursing duration, calf age or cattle breed [28,29]. In contrast, the proportion of time spent sucking, manipulating and pausing during nursing varies widely and seems directly related to milk flow/availability. When calves are getting sufficient milk, very little stripping is observed and most milk release stimulation behaviours (e.g., butting, manipulating, teat changing etc.) occur at low levels [28]. When milk flow/availability is low, calves have been shown to reduce teat fidelity, increase the frequency of teat changes and butting, and have shorter bouts of sucking [25,29,46]. Total sucking and butting are higher at the beginning of nursing, with butting frequency peaking twice, both at the beginning of the meal and following the end of sustained rhythmic sucking [25,28].

Sucking makes up 60–70% and 30–50% of the total nursing time in dairy and beef cattle, respectively (Table 4), with the difference, suggested to be that dairy cattle have higher milk daily yields. Lidfors et al. [25] found that nutritive sucking accounted for less than 35% of the nursing in calves suckling primiparous beef cows. Because primiparous cows generally have lower milk production than multiparous cows, calves of primiparous dams would obtain less milk during nursing and thus spend more time in behaviours that stimulate milk let-down [28]. Thus, it is likely that many of the heifers in the present study had low daily milk production as their calves displayed a high frequency of milk release simulation behaviours, sucked on multiple teats during nursing, with sucking only making up 54% of the total nursing duration.

Similar to dairy calves (73%: [27]; 87%: [28], the beef calves in this study had a preference for suckling the front (64%) rather than the rear teats, which may be due to the front teats being easier to access (Selman 1970b). Unlike dairy calves, the beef calves did not display strong teat fidelity, with all four teats being suckled in 67% of the observed nursings. de Passillé and Rushen [29] demonstrated that dairy calves spend between 40–60% of their nursing time sucking on a single teat, but teat fidelity was less when milk availability was limited. The strong preference for 1–2 d calves to suckle from the LHS of the dam was not an unanticipated result. Selman [24] established that newborn calves typically suckle from one side of the dam only, favouring the side from which they first obtained colostrum. Older calves ($\geq$3 d) did not display this preference similarly, nursing from the LHS (41%) and RHS (55%) of the dam.

## 5. Conclusions

This study has quantified the nursing behaviour of neonatal beef calves and identified dam and calf behaviours that are likely to affect milk intake adversely and potentially survival of beef calves. Calves born to dams that display suboptimal maternal behaviour, frequently terminate nursing bouts or have poor daily milk production are at risk of reduced neonatal growth and may be predisposed to diseases such as omphalitis and infectious enteritis. Based on the observations reported in this study, we propose that the impact of poor or inadequate nursing behaviour is likely to be much more severe in the extensive rangelands of northern Australia, where paddocks are typically 4000–7000 ha in size and distance to watering points are at least 2.5 km. Management strategies which should be considered to address this include selection for maternal behaviour and calving, particularly heifers, in smaller paddocks with easy access to water, shade and adequate pasture.

**Author Contributions:** M.R.M. (study leader), S.D.J. and L.A.H. were responsible for initiating and designing the study. L.A.H. and K.S. were responsible for the detailed planning and execution of the study. A.T.L. and L.A.H. were responsible for data analysis. All authors contributed to the interpretation of the research findings. L.A.H. was responsible for writing the original draft of the paper and M.R.M. was responsible for preparing the final draft of the paper. All authors contributed to the review and editing of drafts of the paper. All authors have read and agreed to the published version of the manuscript.

**Funding:** This research was funded by Meat Livestock Australia, grant number B.NBP.0382.

**Institutional Review Board Statement:** The study was approved by The University of Queensland's Production and Companion Animal Ethics Committee (SVS/346/12).

**Informed Consent Statement:** Not applicable.

**Data Availability Statement:** Substantial research data from this study has been presented in the paper. Access to original data is available on request.

**Acknowledgments:** We thank Rocky Springs Cattle Company for providing access to their farm and cattle, and for logistical support of this research. We also acknowledge C. Petherick (Central Queensland University) and P. Roe (Queensland University of Technology) for their technical assistance in the development of this study's experimental protocols.

**Conflicts of Interest:** The authors declare no conflict of interest. The funders had no role in the design of the study; in the collection, analyses, or interpretation of data, in the writing of the manuscript, or in the decision to publish the results.

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
