# Peer review of "Suckling Behaviour of Beef Calves during the First Five Days Postpartum"

_ruminants, doi:10.3390/ruminants2030022_

Round 1

Reviewer 1 Report

Thank you for your very interesting research!

I have some small suggestions / questions:

Line 54: please add "the" after "towards"

Line 95: was the fortified molasses supplement really given ad lib.?

Line 96: molasses supplement molasses... _ I don't understand, is there a word or a comma missing?

Line 97: you refer to Fig. 1, but it is not clear where in Fig. 1 it is. Maybe where the Trough is? - please specify in the caption to fig. 1

Line 104: replace location by locations (or sites)

Line 130: In table 1 behaviours and definitions of behaviours are not always on the same line / row: it is not easy to identify which one belongs to which; please clarify.

Table 1, row 1: Suckle attempts: how long did they have to  take so they would be assessed as an attempt? Or was each one an attempt, even if it was only for 1 sec.?

Table 1, last row: replace "were" by "where"; replace "stook" by "stood"

Line 131: Table 2, lines 1 - 4: please put the explanations into brackets: (the risk of calf death.... recumbent position.) and (dams should ... after calving.)

Table 2 dam down: >5% of what?

Line 134, Table 3: Dam up: <5% of what?

Line 142: You devidede the data into 12 x 1h-time increments, but on line 114 you said that you observed the animals during 10 h / day, because you didn't do any observations between 13 and 15 h. So is it right to talk about 12 increments? Please explain.

Line 162: replace "or" by "and" (?)

Line 179: maybe better use the same expression as in the methods part: sites or locations

Line 236: please name also the 5 worst

Fig. 4: In graph A please also add the days (1 to 5) in the x-axes. And please increase the distance between graph A and graph B a bit, so it will be clearer.

Lines 284-299: When you are talking about animals showing the same charateristics (like for example on lines 296 or 299) it is confusing that I read in the following brackets P = 0.01 or P = 0.02. I propose to put the remark about the same features after the brackets.

Line 367: you are talking about potential selection traits. This is interesting. Couldn't you add a remark on the heritablity (h2) of such traits? Because for a successful selection we need variability and heritability.

Line 412: add "this" after "pattern". Add "to" after "respect"

Table 4: referring to the reference "Paranhos et al., (2006) you are mentioning 25.0 - 37.8 teat switches per minute. so, that would come to 1 switch every 2 seconds. Is thattrue / realistic? - or is there a mistake?

Lines 454 463: The conclusion does not refer to all the relevant aspects in your findings. Please mention you proposition to select for maternal behaviour. Actually another outcome semms to me that the 1st day is very important and full of risks for a calf; so why not proposing more assistance? for example keep tha pregnant cows in smaller paddocks around calvings where they can be observed? In your last sentence you actually refer to that proposition, but you don't really pronounce it and you hadn't talked about those distances before in your paper. So, I would rather propose to keep them in smaller paddocks around calving than to just state that there is more severeness on those huge paddocks.

General: Maybe an explanation for the expression "teat-strips" would be helpful. I can imagine what it is, but I haven't found a description

Author Response

Many thanks for your helpful suggestions and corrections - please see attachment

Reviewer 2 Report

Article: Suckling Behaviour of Beef Calves During the First Five Days 2 Postpartum by Lindsay A. Hogan et al.

This study reports field observations of 17 heifer-calf pairs over a 5-day period following calving. Cattle were managed in a 6-ha paddock by direct observation and video recording. Their study filled a gap in the literature as, despite substantial amount of work published on mother-young relationship in cattle, very little is known on the suckling behaviour of beef calves during the early neonatal period. The study is sound, the experimental protocol is simple but fits so well in the classic behavioural studies, the manuscript is well structured and easy to read. I have very little comments to make, I really enjoyed reading it. With the exception of some minor clarification, I consider the manuscript perfectly acceptable for publication in Ruminants. Please find below detailed feedback.

 1-     Because the animals were raised outdoors, it would be informative to have some information of the climatic conditions (rainfall, temperature). Some is provided line 97 and also in the discussion line 351, but more precise information in the material and method section would be welcome. Especially since authors did not record the suckling behaviour between 13.00 and 15.00h, partly because it is the hottest part of the day. What was the max and min temperature then? Was the use of the tree cover by heifers at calving due to the temperature? Or is it a hiding behaviour?

2-     Calving was induced by prostaglandin F2α injection. Could this explain partly the difficult birth? No incidence of retained foetal membranes as reported in the literature?

3-     Line 164. “ing rate (sucks/s) was determined as follows number of sucking rates measured/nursing),” Not sure I understand the use of a bracket at the end of the line.

4-     Heifer ranking. Although I did understand the rational of having the ranking system, I was surprised not to have any information on early maternal behaviour: latency to stand up and lick the neonate, licking behaviour, responsiveness (as described in Table 3), potential rejection or mismothering. Some reason is given lines 279-281 but only for suckling behaviour. Also, cattle are considered as being hiders, was this observed in your study? The paper only focuses on suckling behaviour and therefore part of the story is missing. A short paragraph on it in section 3.3 introducing the ranking data would have been welcome. It would also help understand lines 368-372 in the discussion. I suspect that you don’t strictly rely on suckling behaviour to define a “good” or a “poor” mother.

5-     Heifer ranking. Some heifers are ranked as the best or worst 5, but does this rank may have significant impact on calves’ development? In other words, were these differences important or marginal? What would, for instance, be a critical threshold on Figure 2? As it is written that all calves survived until weaning (line 350)

6-     Lines 412-417. I found this section unclear. I wasn’t sure whether growth rate referred to the present work or to the paper cited (Muller et al., 2022). May be writing “they identified two different neonatal growth profile…”. “Neonatal” is misspelled…

Author Response

Many thanks for your helpful suggestions and corrections - please see the attachment
